# Has Food Security and Nutritional Status Improved in Children 1–<10 Years in Two Provinces of South Africa between 1999 (National Food Consumption Survey) and 2018 (Provincial Dietary Intake Study (PDIS))

**DOI:** 10.3390/ijerph19031038

**Published:** 2022-01-18

**Authors:** Nelia P. Steyn, Johanna H. Nel, Linda Drummond, Sonia Malczyk, Marjanne Senekal

**Affiliations:** 1Department of Human Biology, University of Cape Town, Cape Town 7925, South Africa; linda@linda-drummond.com (L.D.); soniamalczyk@gmail.com (S.M.); marjanne.senekal@uct.ac.za (M.S.); 2Department of Logistics, Stellenbosch University, Stellenbosch 7602, South Africa; jhnel@sun.ac.za

**Keywords:** hunger, food security, diet, malnutrition, double burden, non-communicable diseases, stunting

## Abstract

The 1999 National Food Consumption Survey in South Africa showed that food insecurity (hunger) was prevalent in households with children aged one to <10 years. A repeat of the survey in two provinces: Gauteng (GTG) and the Western Cape (WC) was undertaken in 2018. Results showed that in all domains (living areas) in GTG, food shortage prevalence decreased between 1999 and 2018, from 55.0% to 29.6% in urban informal areas, from 34.1% to 19.4% in urban formal areas and from 42.1% to 15.6% in rural areas. While the prevalence of food shortage in urban formal areas in the WC remained similar in 2018, prevalence decreased from 81.8% to 35.7% in urban informal areas and from 38.3% to 20.6% in rural areas. Energy and macronutrient intakes improved significantly in GTG between 1999 and 2018 but not in the WC; intakes were significantly higher in the WC at both time points. The only significant change in stunting, wasting, overweight and obesity prevalence was that 7–<10-year-olds in GTG were significantly more likely to be wasted (BAZ < 2SD) in 2018 than in 1999 (20.2% versus 6.9% respectively). In the WC, 1–3-year-olds were significantly more likely to be obese in 2018 than in 1999 (8.1% versus 1.7% respectively) and 7–<10-year-olds were less likely to be stunted (14.5% versus 4.9% respectively). There were significant negative correlations between the hunger score and dietary variables in both provinces in 1999. In GTG in 2018, only the correlation with fat intake remained while there were still several significant correlations in WC in 2018. Changes in top 12 energy contributors reflect a shift to high or moderate energy foods low in nutrients from 1999 to 2018. Nutrient dense (high micronutrients, low energy/g) foods (e.g., fruit) fell off the list in 2018. Logistic regression analyses reflect the importance for food security of having a parent as head of the household and/or caregiver, and parents having grade 12 or higher education and being employed. We conclude that food security nutritional status indicators improved amongst 1–<10-year-old children especially in GTG between 1999 and 2018. However, the shift to poorer food choices and increase in wasting in older children and overweight in younger children are of concern.

## 1. Introduction

Recent findings on the anthropometric status of children in South Africa indicated that stunting was still highly prevalent (27.4%) in under five-year-old children in 2016 [1]. Furthermore, the prevalence of overweight/obesity (weight-for-height z score (WHZ) > +2SD) was found to be 13.3% overall, 7.2% in boys and 16.4% in girls in 2012 among children 6–9-years-old [2]. One of the major consequences of the double burden of malnutrition is the development of non-communicable diseases (NCDs) in later life [3].

In 1999, the first National Food Consumption Survey (NFCS) was undertaken in South Africa in children 1–9-years-old (*N* = 3120) [4]. Stunting (height-for-age < −2SD) was found to be prevalent nationally (21.6%) and particularly in rural areas (26.5%) while underweight (low weight-for-age < −2SD) was less prevalent nationally (10.3%). Fifty-two percent of children experienced hunger and 23% were at risk of hunger using the Community Child Hunger Identification Project (CCHIP) questionnaire [5] in 1999. Furthermore, significant relationships were found between experiencing hunger or risk of hunger with some anthropometric indicators. Energy and micronutrient intakes at national level were lowest in those who experienced hunger [4].

The South African Nutrition and Health Survey (SANHANES) revealed that in 2012 26% of the population were facing hunger and 28% were at risk of hunger [2]. The latter indicates that many in the population are not food secure even though food balance sheets indicate that nationally South Africa is food secure [6]. This may be due to lack of access to food, lack of availability of food, or lack of stability of access and availability [6]. A report by Oxfam in 2014 revealed that an estimated 13 million people in South Africa faced hunger [7]. They attributed this to numerous factors, including low income and unemployment, gender inequality, lack of access to land, price increases, and poor access to quality and nutritious foods. Statistics South Africa indicated that about 1.6 million households experienced hunger in 2017 with more than 60% of these households being in urban areas. Child hunger was still found to be a challenge in South Africa with more than half a million households with children under five years experiencing hunger in 2017 [6].

Food-insecure children generally have limited access to adequate food because of poverty due to unemployment and lack of access and availability of food, and this may result in an inadequate and poor-quality food intake. Food insecurity may also put children at greater risk of obesity due to increased consumption of lower cost energy-dense-low-nutrient food [8]. Research has shown that in South Africa the rural poor are disproportionately affected by this double burden of malnutrition [9].

To combat poverty and food insecurity, the South African Government has introduced numerous social benefits and programs, including social grants (cash transfers) such as the childcare and foster care grants [10]. Furthermore, a National School Nutrition Program (NSNP) was introduced in 1994 to provide poor children with a daily meal at school [11]; community food garden initiatives and a national fortification program were introduced to combat micronutrient deficiencies [12]. Improvements have also been made to health services with the introduction of the Integrated School Health Program (ISHP) [13] comprising three core components, namely: health assessment and screening, on-site health services, and health promotion. Additionally, improved primary health care clinic services and free health services for children between 0–6 years and expectant and breastfeeding mothers were introduced [14].

These interventions may have contributed to improvements in food security despite the continuing high unemployment and poverty rates since 1994 after the apartheid era ended [14]. There is a paucity of information on changes in food security of households of 1–<10-year-old children and associated socio-demographic predictors since the 1999 NFCS. There is also a paucity of information on changes in relationships between food security and dietary and malnutrition indicators in children over this period.

The aim of the present study was to determine whether food security and nutritional status (diet and anthropometry) have improved in households of 1–<10-year-old children in two provinces since the 1999 NFCS. Furthermore, we aimed to investigate sociodemographic profiles of the households, sociodemographic predictors of food shortage in these households and relationships between household food security and child dietary and malnutrition indicators at each time point.

The study has important implications for policy makers as it provides novel insights in whether food security and nutritional status of 1-year-old to younger than 10-year-old children have improved after introduction of the mentioned initiatives since 1999. This is the only study that has been repeated in the same age group using the same methods over nearly two decades. Results will contribute to identification of the needs for further improvement in food security in the Western Cape and Gauteng.

## 2. Materials and Methods

### 2.1. Sample and Procedures in 1999

The sample of the 1999 NFCS was a nationally representative one including children 1–9-years-old [4]. The sample was drawn using the 1996 census information where an enumerator area (EA) was defined as: “the smallest geographical unit (piece of land) into which the country is divided for enumeration purposes. Enumeration areas contain between 100 to 250 households [15]. In total, 156 EAs were randomly selected from 9 provinces, 82 urban (formal and informal) and 74 non-urban (rural, commercial farms). One child was randomly selected from each household (HH), and a total of 20 children per EA were included in the survey. If there was more than one child present in the prescribed age interval within a HH, then all eligible children in the household in age order were numbered for random selection of one child using a “Random Number Table” designed for this purpose.

The inclusion criteria for the study were as follows: children aged 1–<10-years (12–119 months) old; male or female; availability of a parent/primary caregiver to provide consent; and availability of a parent/primary caregiver to assist with completion of the research questionnaires. The exclusion criteria were as follows: children who were mentally or physically handicapped; children who were on a prescribed diet; children who were ill at the time of the visit or were ill during the past 24 h; children whose mothers/caregivers were unable to respond or appeared to be incapable of responding by providing reliable information; and children whose mother/caregiver was under the influence of alcohol/drugs or was under 15 years old. More detailed information on the sample is available from Labadarios et al. [4].

### 2.2. Sample and Sampling Procedures in 2018 (Provincial Dietary Intake Survey (PDIS)

The sample for the PDIS was selected from two provinces (out of 9 provinces) in South Africa, namely Gauteng (GTG) and the Western Cape province (WC). These provinces showed the highest levels of urbanisation in 2018 and the lowest percentage of household agricultural activities [6]. Three strata (domains) were identified in each of the provinces, namely, urban informal, urban formal, and rural areas for a total of six strata in the overall study. An urban informal area is an unplanned settlement on land that has not been surveyed or proclaimed as residential, and consists mainly of informal dwellings, also referred to as “shacks”. Urban formal areas refer to formal cities and towns characterised by higher population densities, high levels of economic activities and high levels of infrastructure. A rural area (as used in the PDIS) is any area that is not classified as urban, and may comprise tribal areas, farms, or small holdings, and is so designated by Statistics South Africa [6]. The enumerator areas (EAs) were then identified in each stratum and a stratified two-stage sample design was used with a probability proportional to size sampling of EAs at the first stage, and systematic sampling of households within the EAs at the second stage.

The number of households (*N*) per sampling stratum (province and residential area) was calculated to be 175 according to the Household Sampling and Listing Manual [16] as follows:$$N=Deft^{2}\{\frac{\left(\frac{1}{P}-1\right)}{\alpha^{2}}/\left(R_{1}\times R_{2}\times d\right)\}$$
where the design effect (*Deft* = 1.3), the estimated proportion of children classified as stunted (*p* = 0.21), and the desired relative standard error (*α* = 0.2) are based on estimations and guidelines from previous surveys [1]. The proportion (*p*) for stunting was used since the current study formed part of a larger study which also looked at anthropometric status. The individual response rate (*R*_1_ = 0.96) was expected to be higher than the expected household gross response rate (*R*_2_ = 0.89). The number of eligible individuals per household (*d* = 1.06) was calculated as the average number of children aged 1–<10-years per household. It was proposed to survey 175 × 6 strata, or 1050 households.

For the precision of estimates to be acceptable across regions, experience shows that a minimum of 50 dietary interviews per stratum are needed so that reliable estimations for indicators under investigation can be obtained. The final sample allocation reflects a power allocation of 0.5, which is between the proportional allocation and the equal size allocation, so that the survey precision in the urban formal areas is comparable with the urban informal and rural areas, with urban informal and rural areas slightly over-sampled. Since the sample sizes of GTG rural, WC rural and urban informal were less than 150, we increased sampling accordingly to ensure sufficient observations per cell in each age group, with the proposed sample size then being 1050 + 218 = 1268. A total of 84 EAs were selected from the six strata, 25 urban formal, 10 urban informal and 11 rural EAs in GTG, and 18 urban formal, 10 urban informal and 10 rural EAs in the WC. The final sample size was 1326, comprising 733 in GTG and 593 in the WC.

Maps of relevant primary sampling units were generated and passed on to the respective fieldwork teams. An estimate was made of the total number of households (HHs) in each EA to determine the approximate number of qualifying HHs with children within the prescribed age interval in the EA. A listing of eligible households was compiled in all selected EAs, which served as a sampling frame for the selection of households. HHs (a maximum of 16) were then selected based on a predetermined fixed interval (calculated to be specific to each EA) starting from a randomly determined point.

One child in each randomly selected HH was included in the survey. If there was more than one child present in the prescribed age interval within a HH, then all eligible children in the HH in age order were numbered for random selection of one child using a “Random Number Table” designed for this purpose. The inclusion and exclusion criteria for the current study were the same as those in 1999. The final post-hoc stratification weighting reflects the census population of the Western Cape and Gauteng provinces.

In each province, the study was led by a provincial dietitian who was responsible for the overall management of the research team. The field workers were selected based on a minimum level of grade 12 (i.e., completion of high school), as well as other experience in surveys and in field work. Before data collection began, team leaders and field workers received a week-long extensive training session, according to a manual which had been developed for the purpose of the study, facilitated by experienced researchers in anthropometric measurements, as well as the delivery of sociodemographic questionnaires and other questionnaires. The training session included standardising the anthropometric measurements done by the field workers against a trained and experienced anthropometric researcher.

### 2.3. Measures

#### 2.3.1. Sociodemographic Profile and Hunger Scale Questionnaire

Only sociodemographic variables that were available at both time points were included in the present study. These included who looks after the child most of the time, head of the household, marital status of the mother, mother’s highest level of education and employment status, father’s employment status and area of residence. Data on grants, household income and school feeding participation were not available.

Food insecurity as indicated by hunger was measured using the Community Childhood Hunger Identification Project (CCHIP) questionnaire [5] in both 1999 and 2018. This questionnaire measures household, child, and individual level food security. Altogether, there are eight questions in the scale. If any of these are affirmative, then a score of one is given. A total score of 5–8 indicates that a food shortage (food insecurity) is present in the household. A score of 1–4 indicates that the household is at risk of hunger (poor food security) and a score of zero indicates that the household is food secure. These scores were used to calculate an association with selected dietary and anthropometric variables.

#### 2.3.2. Dietary Intake

In 1999, dietary intake was assessed using a single 24 h recall. Fieldworkers were trained by dietitians in each province [4]. Dietary intake was assessed by interviewing the primary caregiver of the child. Food models were used to assess portion sizes eaten. Week and weekend days were covered proportionally. Dietitians checked the questionnaires and coding was done at a central station under supervision of a statistician.

A single 24 h recall was also used in 2018, but two additional recalls were completed in a sub-sample. All dietary interviews took place in the presence and with the input of the mother/primary caregiver. For 1–5-year-old children, the mother/caregiver reported on the intake of the child on the previous day, with no input from the child. For 6–9-year-old children the mother/caregiver and child were interviewed together, to record the dietary intake during the prior 24 h. If the child had been at a day care center on the previous day, the day care center was visited by the fieldworker to determine dietary intake.

The multiple pass method of the 24 h recall was used to administer the 24 h recall [17]. Essentially, the interviewer first went through the previous day’s intake by recording all the food items and drinks that were consumed between waking up in the morning until going to sleep in the evening (and during the night if applicable). Recall was helped by the interviewer going through the daily activities with the participant and linking them to eating occasions. Next, the interviewer prompted the respondent to identify food and drinks that may have been “forgotten,” such as cold drinks, candies and snacks. Information was then recorded regarding when and where the various food items were consumed. Following this, more detailed information was obtained regarding the preparation of the foods and individual ingredients as relevant.

Dietary intakes were calculated based on the South African Food Composition Tables [18]. Portion sizes were obtained using a booklet adapted from the Dietary Assessment and Education Kit (DAEK) [19]. The booklet comprises life size sketches of generic household utensils and crockery and life size portions of actual foods e.g., different slices of bread varying in size and thickness, to make estimations of portion size as accurately as possible. Generic three-dimensional food models made from flour were also used to assist in recording volume measures such as porridge and rice.

For investigation of changes in quality of food intake, the top 12 contributors to total energy intake in each province at each time point were identified. These food items were classified as: (1) Energy moderate/dense and nutrient poor (reflecting the poorest quality food items), or (2) Energy moderate/dense and nutrient rich, or (3) Nutrient dense (high nutrients, low energy density).

#### 2.3.3. Anthropometry of Children

In 1999 and 2018, electronic digital scales were used by trained fieldworkers to determine weight in accordance with standard procedures. The scales were calibrated regularly using standard weights. The children were weighed in light clothing (without coats, cardigans, and shoes) and the reading was recorded to the nearest 100 g. The measure was then repeated and the average of the two readings used [20].

The height of children older than two years was measured in 1999 and in 2018 without shoes using a portable stadiometer, which was placed on an even surface. Children and mothers stood on the base board with their backs to the vertical rod of the stadiometer, facing the fieldworker. Children stood upright with shoulders relaxed and heels against the measuring board, arms hung loosely by the side and the head in the Frankfurt plane. The fieldworker then lowered the headboard until it touched the head (any hairclips/pieces that may have impacted the reading were removed prior to taking the measurement). The reading was taken to the nearest 0.1 cm. The measurement was then repeated, and the average used [20].

Children who were under 24 months old had their supine length measured on a measuring mat (SECA 210 mobile measuring mat for babies and toddlers, SECA, Hamburg, Germany). This involved the fieldworker and the mother/primary caregiver placing the child on his/her back on the board. The crown of the head touched the top of the headboard. The mother/primary caregiver then held the child in a flat position with legs straight while the footboard was moved up to touch the heels. Length was measured to the nearest 0.1 cm. The measurement was repeated twice, and the average used [20].

The prevalence of stunting, underweight, wasting, overweight and obesity were determined based on the WHO growth standards [21,22]. The WHO growth standards were used to calculate height-for-age z scores (HAZ), weight-for-age z scores (WAZ) and body mass index z scores (BAZ). Stunting was determined as HAZ < −2SD, wasting as BAZ <−2SD, overweight as BAZ > +2SD but ≤ 3SD for under 5-year-olds and BAZ > +1SD but ≤ 2SD for 5 year and older children and obese as BAZ > +3SD for under 5-year-olds and BAZ > +2SD for 5 year and older children.

#### 2.3.4. Data Analyses

Data analyses were conducted using SAS Version 9.4, SAS for Windows (SAS Institute, Cary, NC, USA). Weighted means, proportions and 95% confidence intervals were calculated by incorporating the strata and cluster structures in the data.

All analyses were performed by taking the complex survey designs into consideration, this is, for both the NFCS 1999 and the PDIS 2018 surveys. Sample weights were used. Relationships for categorical variables with province, GTG and the WC, and time point, 1999 and 2018, were tested for significance using the Rao–Scott Chi-Square test. Weighted means, medians, standard errors and confidence intervals for macronutrient intakes (non-normally distributed), z-scores (normally distributed), as well as the total hunger score (count variable), were calculated, and significant differences between provinces on the one hand, and between surveys on the other hand, were calculated using the LSMEANS statement in PROC SURVEYREG, to execute the independent *t*-test for (normally distributed) and the Wilcoxon two-sample test for non-normally distributed and count variables.

Associations between the hunger score and dietary (total energy-, protein-, fat- and carbohydrate intakes) and anthropometric variables (HAZ and BAZ) were investigated using the Spearman correlation coefficient.

Sociodemographic predictors of having a food shortage (dependent variable) were determined with bivariate logistic regression analyses using PROC SURVEYLOGISTIC. Odds ratios and 95% confidence intervals are reported for each independent variable for each of the two provinces at each of the two timepoints.

For energy and macronutrient intake analyses the single 24 h recall completed in the NFCS was compared with the single 24 h recall completed by the total sample in the PDIS. Although the PDIS data was adjusted to remove within-person variance (using the National Cancer Institute method) [23] including two additional recalls in a subsample of 11% of the total sample, reflecting usual intake, the NFCS data could not be similarly adjusted to obtain usual intake due to a lack of repeated recalls. As a result, only results of the single 24 h recalls were compared.

## 3. Results

The sociodemographic profiles of households of children who participated in the NFCS and PDIS for given variables are presented in Table 1. No significant relationships were found within GTG or the WC for who looks after the child most of the time in 1999 and 2018 (mostly mother/father in both provinces at both timepoints) or for the mother’s highest level of education (just under or just more than 50% in both provinces at both time points had an education level less than matric). Head of the household within GTG only was mostly the father/grandparent at both time points. Significant relationships were found within GTG and WC for marital status of mothers in 1999 and 2018 (mothers were less likely to be married in 2018 than in 1999 in both provinces), employment status of the mother (mothers were more likely to be employed in 2018 than in 1999 in both provinces), employment status of the father (fathers were less likely to be employed in 2018 than in 1999 in both provinces), area of residence (households were less likely to be in urban informal or rural areas in 2018 than in 1999 in both provinces), as well as head of the household in the WC only (fathers were less likely to be head of the household in 2018 than in 1999 in the WC).

Significant relationships were found between province and marital status of mothers (mothers in GTG were less likely to be married than mothers in the WC in 1999 and in 2018) and employment status of mothers (mothers in GTG were more likely to be employed than those in the WC in 1999 and in 2018) (statistics in footnote to Table 1).

Table 2 shows that there were no significant relationships between food security categories and time points (1999 vs. 2018) in the total group of children or the three age groups in the WC. However, significant relationships were found in GTG. Households in this province were less likely to experience food shortage in the total group and in the youngest and oldest age groups (1999 vs. 2018): total group: 41.0% vs 20.2%; 1–3-year-olds: 45.5% vs. 18%; and 7–<10-year-olds: 41.6% vs 18%). The mean hunger score in GTG decreased significantly from 3.4 in GTG in 1999 to 1.9 in 2018 (*p* < 0.001), while the decrease in the Western Cape was from 2.6 in 1999 to 2.2 in 2018 (*p* < 0.05).

Table 3 shows that all the responses to the CCHIP questions asked in 1999 improved significantly in 2018 in Gauteng; in some cases, by about 40–50%. Questions 1–4 relate mainly to access of food while 5–8 relate to child hunger. For example, Question 5: “Do your children ever eat less than you feel they should because there is not enough money for food?” This question was answered yes by 40.7% respondents in GTG in 1999 and by 20.5% in 2018 (nearly 50% decrease). This trend was followed for all questions in GTG but not in WC, where only one question showed a significant change (Question 8: “Do any of your children ever go to bed hungry because there is not enough money to buy food?”) This question was answered yes by 13.5% in 1999 and by 5.6% in 2018. The “no risk” category increased from 37.8% in GTG in 1999 to 57.3% in 2018 (*p* < 0.001). The change in the WC was not significant.

In 1999, 55.0% of respondents in GTG urban informal areas were categorised as experiencing a food shortage, while this was 42.1% in rural areas and 34.1% in urban formal areas (Table 4, Figure 1). In all domains percentages reporting a food shortage decreased in 2018, to 29.6% in urban informal areas from 55%, to 19.4% in urban formal areas from 34.1% and to 15.6% in rural areas from 42.1%. In 1999 in the WC, households in urban informal areas were significantly more likely to experience food shortage (81.8%) than those in rural (38.3%) and urban formal areas (20.2%) (Table 4, Figure 2). While the percentage in urban formal areas with a food shortage remained similar in 2018, it decreased to 35.7% in urban informal areas from 81.8% and to 20.6% from 38.3% in rural areas.

Table 5 shows significant increases in energy and macronutrient intakes between 1999 and 2018 in GTG. Only median protein values in the two older age groups did not increase significantly between 1999 and 2018. In the WC, median intakes of energy and macronutrients did not change significantly between 1999 and 2018 in most age groups. However, median carbohydrate intake decreased significantly from 1999 to 2018 in the oldest group, as did median energy intake in the 3–6-year-olds.

Energy, total protein, total fat and carbohydrate intakes were significantly higher in the WC than GTG (*p* < 0.01) in 1999 in all age groups. In 2018, this was also true for energy, total protein and fat intakes in 1−<3-year-olds (*p* < 0.01), for total protein (*p* < 0.05) and total fat (*p* < 0.01) in 3–6-year-olds and for total protein and carbohydrate intakes in 7−<10-year-olds.

Table 6 provides perspectives on changes in the quality (nutrient density) of the top 12 food items contributing to the total energy intake of 1–<10-year-old children in each province at each time point. Bread (white and brown) and maize porridge were not yet fortified in 1999 and thus classified as moderate energy and nutrient poor for that time point. For 2018, these items were classified as moderate energy and nutrient rich because of the fortification with iron, zinc, vitamin A, thiamine, riboflavin, niacin, vitamin B6 and folate since 2003 [12]. Nutrient dense items such as fruits (other): e.g., apples, pears, bananas) vitamin C-rich vegetables (e.g., cole slaw, tomatoes, broccoli) that were among the top 12 in 1999 fell away in 2018 and in both provinces the number of ‘energy dense nutrient poor or moderate energy nutrient poor’ items in the top 12 increased, even though bread and maize porridge were categorised as ‘moderate energy nutrient rich’ instead of ‘moderate energy nutrient poor’ in 2018. Items that stand out in 2018 include salty snacks (not in the top 12 in 1999 in either province), potato with fat (only in top 12 in WC in 1999 but moved from 12th to 4th position in terms of contribution to total energy intake from 1999 to 2018), commercially processed meats (not in the top 12 in 1999 in either province) and inclusion of sweets in the top 12 in GTG in 2018 and pasta in the WC. Sugar and white rice were in the top 12 in both provinces at both time points. The type of fat in the top 12 energy contributors changed from hard margarine in both provinces in 1999 to polyunsaturated oil in GTG and polyunsaturated fat in WC in 2018.

The only significant change in stunting, wasting, overweight and obesity prevalence was that 7–<10-year-olds in GTG were significantly more likely to be wasted (BAZ < −2SD) in 2018 than in 1999 (20.2% versus 6.9% respectively) (Table 7). In the WC, 1–3-year-olds were significantly more likely to be obese (BMI > +3SD) in 2018 than in 1999 (8.1% versus 1.7% respectively) and 7–<10-year-olds were less likely to be stunted (14.5% versus 4.9% respectively).

Table 8 presents Spearman correlation coefficients between the total hunger score and dietary and anthropometric indicators in the total sample of 1–<10-year-old children. There were significant negative correlations between the hunger score and total energy-, protein-, fat- and carbohydrate intakes and between the hunger score and HAZ and BAZ in both provinces in 1999, with the exception of BAZ in GTG in 1999. The only significant correlation that remained in GTG in 2018 was with total fat intake (negative correlation) indicative of improvements in GTG, while negative correlations between the hunger score and total energy, protein and fat intake, as well as with HAZ remained for the sample of children in the WC.

Predictors of and protectors against experiencing a food shortage are depicted in Table 9. Factors that increased risk of food shortage in GTG in 1999 included having a grandparent as head of the household, having an unmarried mother and living in an urban informal area. Protectors against food shortage in GTG in 1999 were being 4–6 years old, having a mother with a grade 12 or a post grade 12 education, and having an employed mother (education of the father was not assessed in 1999). There were no risk predictors in GTG in 2018. In 1999 in the WC, predictors of risk of food shortage were having a caregiver other than the mother, the head of the household not being the mother or the father or a grandparent i.e., a sibling, aunt or uncle, having an unmarried mother and living in an urban informal or rural area. Protectors against food shortage in the WC in 1999 were having a mother with a grade 12 or a post grade 12 education, having an employed mother and having an employed father. In the WC in 2018, only living in an informal urban area was found to be a predictor of risk of food shortage. Having a father with a post-grade 12 qualification, having an employed mother and having an employed father were protectors against food shortage in the WC in 2018.

## 4. Discussion

In this research we set out to compare food security and sociodemographic predictors thereof, as well as dietary and anthropometric indicators of 1–<10-year-old children living in two South African provinces between the 1999 NFCS [4] and the 2018 PDIS [24,25]. In summary, there were improvements in food security in both provinces, apart from urban formal areas in the WC, where there was little change. Dietary intake indicators (significant increases in energy and macronutrient intakes) reflect the improved food security in GTG; as do the anthropometric indicators in both provinces. Regarding sociodemographic variables, there were no specific factors to account for improvements in food security except that there was a significant increase in urban formal residency in both provinces between 1999 and 2018. This is likely to be related to the large increase in formal housing as part of the ANC Government’s Reconstruction and Development Program introduced after 1994 [14]. Earlier research has shown that the highest prevalence of stunting was found in rural and informal areas [4]. Improvements were also found in correlations between the hunger score and dietary and anthropometric indicators between the two time points, particularly in GTG.

One of the detrimental anthropometric outcomes in both provinces was, however, the increase in the prevalence of overweight and obesity, particularly in the 1–3-year-old group. This is also the group that showed the highest stunting with 39% of the one-year-olds and 23.2% of the two-year-olds having a HAZ < −2SD in both provinces. This decreased to 17.2% in the three-year-olds and to 11.9% in the 4-year-olds [24]. The finding that both stunting and overweight/obesity are prevalent in the 1–3-year-olds is of some concern since it is proposed that a child’s linear growth potential is largely determined by the time that they are two years old (within the 1000 days window); thus, chronic poor nutrition during the first 1000 days of life may result in stunting and as a result negatively affect future growth and health with stunted children accumulating a greater fat mass than their normal counterparts [26]. Overall wasting increased from 10.3% to 14.4% in GTG and decreased from 11.8 to 10.1% in WC. It is particularly high in older children (7–<10 years) and is difficult to explain since poor primary school children receive school meals in the NSNP.

Numerous factors may have been responsible for the significant improvements in food security in GTG and in WC. Energy intake increased significantly in GTG which may explain improvement in growth. However, many of the commonly consumed items are energy-dense, such as sugar sweetened beverages, candies, and salty snacks [25]. Another factor which may have contributed to this is the availability of social grants from the Department of Social Services who introduced numerous grants since 1994 based on the high levels of unemployment, and the declining household agricultural production in rural areas [27]. Poverty dictates household income and access and availability to nourishing food. Protectors in GTG in 2018 were having a mother with a grade 12 education, having a father with a grade 12 or post grade 12 education, having an employed mother and/or having an employed father. In WC in 2018, these were having an employed mother and/or father.

The following social grants are currently available in South Africa: child support grant; care disability grant, disability grant, foster care grant, older persons grant, social relief of distress, and war veterans grant [28]. In 2018, 17 million individuals received social grants in the form of cash transfers. The most common one was the child support grant which was paid to 12.1 million primary caregivers of children under 18 years old, 3.3 million old age pensioners above the age of 60 years and 1.1 million people living with disabilities [27].

Overall, numerous studies which have evaluated the benefits of social grants have shown positive outcomes in terms of reducing food insecurity [29,30,31,32,33]. Data from the Income and Expenditure Survey data of 2010/2011 were analysed and it was found that social grants are well targeted; more importantly, they have significantly reduced poverty levels [29]. These results include those in female-headed households, rural areas, the African population, and specifically in the Eastern Cape and Limpopo provinces which have high poverty levels.

Case et al. [32] used longitudinal data from the Africa Centre for Health and Population Studies to evaluate the Child Support grant. This grant appears to be reaching children living in the poorest households. Furthermore, children who received the grant were significantly more likely to be enrolled in school than equally poor children of the same age. Thus, the grant “appears to overcome the impact of poverty on school enrolment”. Delany et al. [33] also evaluated the benefits of the Child Support grant. They discussed several reasons why this grant has led to a decline in poverty among South African children with two out of three children benefiting from it. Some of the benefits include the fact that the grant also extends to older children and the grant amount although modest is revised yearly. The grant is well targeted, with the bulk of spending on the grant going to the poor. It has been found that the child support grant is mainly accessed by informal sector working women. This means that they have access to income in the period following birth. This is a very important since they have access to money to purchase food and it also allows them to stay with the infant to breastfeed them instead of returning to work which is critically important in the prevention of stunting [34,35].

The NSNP aims to enhance the learning capacity of learners through the provision of a healthy meal at schools. Where it is implemented, the programme has been shown to improve punctuality, regular school attendance, concentration and the general wellbeing of participating learners. Whilst learners are being provided with nutritious meals, they are also taught to establish and maintain good eating and lifestyle habits for life. Numerous studies have indicated that many learners do not eat breakfast before coming to school and this was one of the drivers of the NSNP [11]. Faber et al. examined the NSNP at 90 purposively selected poorly resourced schools in South Africa [36]. The study found that schools did not comply with the mandate of serving vegetables and/or fruits everyday as stipulated by the Department of Education. Furthermore, many schools have school stores which frequently sell unhealthy food items such as salty snacks, crisps, candies and sugar sweetened beverages. This may have contributed to the increased prevalence of overweight found in children.

Migration is another factor which may impact on food insecurity. A Statistics South Africa report shows that South Africa is estimated to receive a net immigration of 1.02 million people between 2016 and 2021 [37]. Most international migrants settle in GTG (47.5%) by virtue of what is known as the “pull factor” [6]. Since GTG is the economic hub of the country, it attracts international migrants as well as domestic migrants from rural provinces such as Limpopo and Eastern Cape. They move because they are hoping for jobs and a better quality of life. Coupled with improved government grants and services provided by Government, GTG has shown significant improvements. This may explain why food security has improved significantly in GTG, including as regards child hunger, as shown in the results.

The WC received the second highest number of in-migrants for the period 2016 to 2021. This is the result of “push factors” which drive people from the Eastern Cape towards the WC due to lack of job opportunities and poor economic conditions in the Eastern Cape [6]. During the first quarter of 2018, the Eastern Cape had the highest unemployment rate in the country at 35.6% while the national figure was 27.1%. This situation is thought to be responsible for the finding that there were less improvements in WC compared with GTG. Additionally, severe drought in 2018 in WC further exacerbated the situation [6].

According to Ronquest-Ross and colleagues, large food consumption changes have taken place since 1994 after the end of apartheid [38]. Their findings indicate that food consumption shifts have been towards an overall increase in daily kilojoules consumed, an increase in the proportion of processed and packaged foods and edible vegetable oils, an increase in sugar sweetened beverages, an increase in added sugar, an increase in intake of animal-source foods, and a move away from fruit and vegetables. These shifts in food consumption are concerning as they relate to the kilojoule, sugar, fat, and salt composition of foods and the potential knock-on effects of these nutrients on public health and the double burden of malnutrition.

Another factor influencing food security is the distance between where people live and the nearest food source, as well as their ability to travel that distance. Food distribution at household level also impacts food security where one or more household members may allocate their food to other household members such as children, with parents sheltering their children against food insecurity. In South African rural areas, households may not have access to healthy foods such as fruit and vegetables and whole grain foods due to the distance to shops, and these items may not always be available in these areas. This is also a likely scenario in urban informal areas where healthier foods may only be available at additional cost or not available at all [39]. This has been shown by the finding that living in informal areas was a significant predictor of hunger in both provinces.

The quality of the diet of children from low-income backgrounds in South Africa has been found to be generally poor, in terms of being energy-dense, high in sugar, high in sodium, and low in fruit and vegetables [25,40,41]. Much unhealthy snacking takes place among children [42,43]. This type of diet can lead to the development of obesity and non-communicable diseases (NCDs) in later life [44]

Farrell and colleagues [45] have reviewed the relationship between food insecurity and obesity in low- and middle-income countries. The review generated 13 peer-reviewed articles and the affordability of high-energy, processed foods was identified as a main mechanism, which determined whether food insecurity leads to obesity. Food prices and levels of income impact on this. According to Farrell et al. [45], studies most likely to show a relationship between food insecurity and obesity were those where energy-dense commercial foods were available at low cost. In South Africa, it has been shown that refined maize meal is the staple food in the black African population and there is a higher intake of refined maize meal consumption in GTG than in WC. Consumption of white bread is the second most common high-energy commercial food which is also commonly consumed in both provinces although to a greater extent in the WC [24,25,46].

Mothers’ level of education remains an important predictor of food insecurity as indicated by the findings of this study. Key findings from a study in the USA which compared children from mothers who had not completed high school compared with those who had a bachelor’s degree, showed the following health disparities: 9.0% vs. 6.8% for low birthweight, 8.2% vs. 3.9% deaths to children under age 1 per 1000 live births, and 27% vs. 13% for obesity [47]

Eisenmann et al. [48] evaluated the relationship between food security and obesity. The results were varied with positive, negative and no associations. The reasons for the mixed results are difficult to disentangle. Nevertheless, according to them, all the studies to date have shown that obesity and food security co-exist, whether the relationship is significant or not.

A limitation of the research is that only two provinces were studied so it is not possible to generalise the results to all nine provinces. Furthermore, mothers/caregivers may have answered the hunger scale questions in the hope of receiving assistance from government if they plead hunger. A further limitation is the fact that the study took place in 2018 before COVID-19 so would not reflect the impact of that.

## 5. Conclusions

We conclude that the results of this study show that overall, the food security and anthropometry of 1–<10-year-old children in the WC and GTG improved since 1999. In contrast with 1999, no predictors of risk of food security remained in GTG and only one in the WC (living in an informal area). Protectors against food shortage remained similar over time and mostly involved, having better educated and employed parents. The improvement in food security may have contributed to the increase in energy intake of children in GTG and subsequent improvements in nutritional status. However, it is a concern that the improvements in food security have been accompanied by increased overweight and obesity, especially in the younger children in both provinces.

In terms of policy formulation, the results of this research imply that the various initiatives introduced by the South African government may have had the necessary effect. However, further research is essential to confirm the contribution and-cost-effectiveness of the different initiatives to identify those that should be continued into the future and those that should/could be phased out. The shift seen in food choices to more energy dense low nutrient items is a concern and should be taken up as a priority for intervention initiatives by local and national departments of health in the country.

## Figures and Tables

**Figure 1 ijerph-19-01038-f001:**
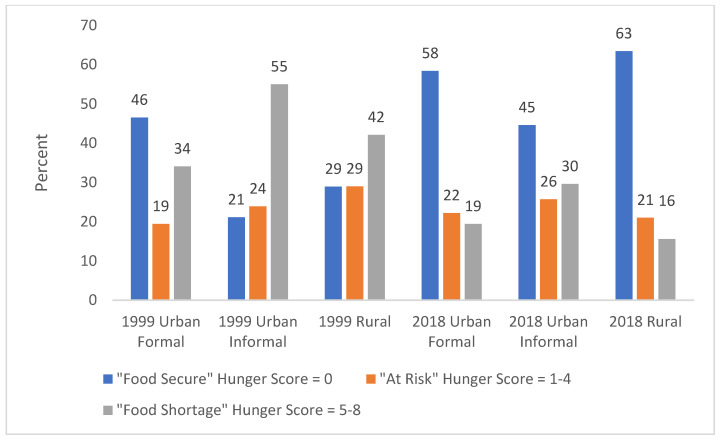
Percent households with different hunger scale categories in Gauteng (values rounded to nearest whole number).

**Figure 2 ijerph-19-01038-f002:**
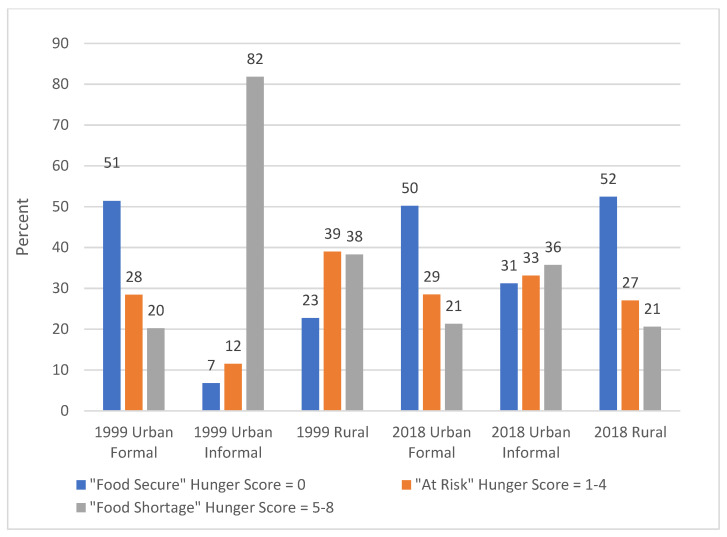
Percent households in Western Cape with different hunger scale categories (values rounded to nearest whole number).

**Table 1 ijerph-19-01038-t001:** Socio-demographic profiles ^1^ of 1–<10-year-old children in Gauteng and Western Cape, in 1999 and 2018 and relationships with timepoint within provinces and between provinces at a particular timepoint.

Sociodemographic Variables	Gauteng Weighted Percentage (Standard Error)			Western Cape Weighted Percentage (Standard Error)		
	1999 ^2^	2018 ^3^	Rao-Scott Chi-Square *p* Value *	1999 ^2^	2018 ^3^	Rao-Scott Chi-Square *p* Value *
	*N* = 418	*N* = 733		*N* = 353	*N* = 592	
Who looks after the child most of the time						
Mother/Father	75.1 (4.4)	76.7 (2.3)	0.889	78.4 (4.8)	72.7 (3.1)	0.526
Grandparent	17.4 (3.5)	16.9 (2.0)		16.8 (3.9)	21.1 (2.7)	
Other (sibling, aunt, uncle)	7.5 (1.7)	6.4 (1.4)		4.9 (1.3)	6.2 (1.9	
Female	53.1 (2.8)	50.1 (2.4)		51.5 (4.1)	52.4 (2.2)	
Head of household						
Father	48.3 (5.6)	40.5 (3.2)	0.117	59.7 (6.2)	39.1 (2.1)	0.001 **
Mother	15.4 (2.4)	17.2 (1.5)		9.9 (2.2)	10.8 (1.9)	
Grandparent	33.3 (6.6)	33.2 (3.8)		26.7 (5.2)	41.7 (3.0)	
Other (e.g., aunt, uncle, friend)	3.1 (0.8)	9.1 (1.9)		3.7 (1.1)	8.4 (1.6)	
Marital status of mother						
Married	41.6 (5.0)	24.3 (2.2)	0.001 **	62.5 (5.4)	41.2 (3.9)	0.002 **
Other	58.4 (5.0)	75.7 (2.2)		37.5 (5.4)	58.9 (3.9)	
Mother’s highest education						
Less than matric	58.2 (4.9)	54.2 (3.0)	0.785	48.9 (6.4)	59.8 (5.3)	0.323
Matric ^#^	30.7 (3.1)	34.2 (2.9)		33.2 (3.1)	24.2 (3.5)	
Qualification after matric	11.2 (4.0)	11.6 (1.8)		17.8 (5.2)	16.0 (4.0)	
Mother’s employment status						
No/don’t know/NA	30.7 (5.3)	22.2 (2.3)	0.120	46.6 (4.5)	39.3 (3.7)	0.207
Yes	69.3 (5.3)	77.8 (2.3)		53.4 (4.5)	60.7 (3.7)	
Father’s employment status						
No/Don’t know/NA	50.0 (3.7)	64.2 (2.2)	0.001 **	56.3 (2.5)	65.3 (2.8)	0.019 *
Yes	50.0 (3.7)	35.8 (2.2)		43.7 (2.5)	34.7 (2.8)	
Type of residence						
Urban formal	64.8 (1.6)	88.5 (0.4)	<0.001 ***	78.8 (0.7)	86.9 (0.5)	<0.001 ***
Urban informal	31.7 (1.6)	9.0 (0.3)		11.0 (0.7)	6.6 (0.2)	
Rural	3.4 (0.4)	2.5 (0.1)		10.2 (0.3)	6.6 (0.5)	

NA = Not Applicable. ^1^ Only socio-demographic indicators that were available at both time points (1999 and 2018); four profiles in total, one for each province at each time point. * Rao–Scott Chi-Square test; comparison within each province in 1999 and 2018 values of sociodemographic variables were adjusted using relevant weighting. Frequencies performed by incorporating weights and complex survey design. * *p* < 0.05; ** *p* < 0.01; *** *p* < 0.001. ^#^ The category “Matric” represents passed grade 11 or grade 12 in 1999, but only grade 12 in 2018. ^2^ Rao-Scott Chi-Square test; comparison between provinces in 1999; relationships: marital status of mother (*p* < 0.001) and employment status of mother (*p* < 0.05). Not shown in the table. ^3^ Rao–Scott Chi-Square test; comparison between provinces in 2018; relationships: marital status of mother (*p* < 0.001) and employment status of mother (*p* < 0.001). Not shown in the table.

**Table 2 ijerph-19-01038-t002:** Comparison of food security (hunger) of households in the Western Cape and Gauteng in 1999 and 2018 overall and by age group.

Level of Food Security	Gauteng Weighted Percentage (95% CI)			Western Cape Weighted Percentage (95% CI)		
Hunger ^$^	1999 *N* = 408	2018 *N* = 717	Rao-Scott Chi-Square *p* Value	1999 *N* = 342	2018 *N* = 592	Rao-Scott Chi-Square *p* Value
% No risk (95%CI)	37.8 (27.5–48.2)	57.3 (48.8–65.7)	<0.001 ***	43.6 (27.4–59.7)	49.1 (39.6–58.6)	0.505
% At risk (95%CI)	21.2 (16.8–25.5)	22.5 (17.5–27.5)		27.6 (18.9–36.3)	28.7 (23.0–34.5)	
% Food shortage (95%CI)	41.0 (31.6–50.3)	20.2 (15.0–25.5)		28.8 (20.1–37.5)	22.2 (16.7–27.7)	
By age group:	*N* = 230	*N* = 288		*N* = 155	*N* = 240	
% 1–3 years No risk (95%CI)	36.9 (28.7–45.1)	57.0 (47.1–66.9)	<0.001 ***	34.7 (21.7–47.7)	43.2 (30.7–55.7)	0.283
At risk (95%CI)	17.6 (11.4–23.8)	25.0 (17.9–32.1)		30.3 (20.3–40.3)	33.1 (23.7–42.4)	
Food Shortage (95%CI)	45.5 (37.2–53.9)	18.0 (11.5–24.6)		35.0 (25.2–44.8)	23.8 (12.9–34.6)	
By age group	*N* = 129	*N* = 264		*N* = 129	*N* = 197	
% 4–6 years No risk 95%CI	39.3 (25.0–53.6)	57.1 (47.2–67.0)	0.039 *	50.1 (31.3–69.0)	54.3 (44.3–64.2)	0.481
At risk (95%CI)	24.6 (16.1–33.1)	18.2 13.7–22.6)		19.7 (10.4–29.1)	24.6 (16.1–33.1)	
Food shortage (95%CI)	36.1 (25.3–46.8)	24.7 17.2–32.2)		30.2 (13.7–46.7)	21.2 (13.0–29.3)	
By age group	*N* = 49	*N* = 165		*N* = 58	*N* = 155	
% 7–<10 years No risk (95%CI)	37.1 (21.1–53.1)	57.8 (47.3–68.3)	0.011 *	46.4 (24.2–68.7)	50.7 (38.2–63.2)	0.596
At risk (95%CI)	21.3 (12.1–30.5)	24.2 15.1–33.3)		36.8 (18.1–55.6)	27.9 (18.9–36.8)	
Food shortage (95%CI)	41.6 (25.5–57.7)	18.0 10.0–26.1)		16.7 (5.7–27.8)	21.4 (15.1–27.7)	
Hunger Score	*N* = 408	*N* = 717		*N* = 342	*N* = 592	
Mean	3.4	1.9	<0.001 ^###^	2.6	2.2	0.026 ^#^
95% CI of mean	(2.8–4.1)	(1.5–2.3)		(1.8–3.4)	(1.7–2.6)	
Median	2.6	0.0		0.7	0.1	
95% CI of median	(0.7–4.5)	(0.0–0.6)		(0.0–1.7	(0.0–0.7)	

^$^ Community Child Hunger Identification Project (CCHIP) questionnaire [5]. Hunger score categories: No risk (hunger score = 0); at risk (1 ≤ hunger score ≤ 4); food shortage (5 ≤ hunger score ≤ 8). 95th CI = 95th percent confidence interval. * *p* < 0.05; *** *p* < 0.001: Significant relationships within provinces comparing 1999 with 2018, Rao–Scott Chi-Square test. Means calculated by incorporating weights and complex survey designs; ^#^ *p* < 0.05; ^###^ *p* < 0.001: Significant difference within provinces of GTG and WC hunger scores in 1999 and 2018, Wilcoxon two-sample test.

**Table 3 ijerph-19-01038-t003:** Comparison of results for individual questions on the hunger questionnaire ^$^ between households of 1–<10-year-old children in Gauteng and Western Cape provinces in 1999 and 2018.

Questions on the CCHIP ^$^ Questionnaire	Gauteng			Western Cape		
	% Yes (95%CI): 1999 (*n* = 408)	% Yes (95%CI): 2018 (*n* = 717)	Rao-Scott Chi-Square *p* Value	% Yes (95%CI): 1999 (*n* = 342)	% Yes (95%CI): 2018 (*n* = 592)	Rao-Scott Chi-Square *p* Value
1. Does your household ever run out of money to buy food?	55.0 (45.4–64.5)	36.7 (28.8–44.7)	0.003 **	46.5 (32.1–60.9)	46.2 (37.4–55.1)	0.9723
2. Do you ever rely on a limited number of foods to feed your children because you are running out of money to buy food for a meal?	51.3 (40.4–62.3)	32.9 (24.9–40.9)	0.006 **	44.0 (30.6–57.3)	39.5 (30.6–48.4)	0.556
3. Do you ever cut the size of your meals or skip any because there is not enough food in the house?	44.5 (36.2–52.8)	25.8 (18.8–32.7)	0.006 **	37.6 (25.3–49.9)	32.1 (24.0–40.3)	0.435
4. Do you ever eat less than you should because there is not enough money for food?	48.6 (39.6–57.6)	27.1 (20.2–34.0)	0.002 **	39.4 (26.9–51.8)	33.5 (25.4–41.7)	0.410
5. Do your children ever eat less than you feel they should because there is not enough money for food?	40.7 (32.1–49.4)	20.5 (15.5–25.6)	<0.001 ***	29.2 (19.2–39.2)	22.8 (15.8–29.8)	0.268
6. Do your children ever say they are hungry because there is not enough food in the house?	37.3 (28.8–45.8)	20.1 (14.6–25.5)	0.03 **	23.8 (15.9–31.6)	17.9 (12.4–23.5)	0.199
7. Do you ever cut the size of your children’s meals, or do they ever skip meals because there is not enough money to buy food?	37.6 (29.3–45.8)	17.6 (12.9–22.3)	<0.001 ***	25.2 (16.9–33.5)	17.7 (12.6–22.9)	0.101
8. Do any of your children ever go to bed hungry because there is not enough money to buy food?	27.8 ^#^ (21.0–34.6)	10.0 (6.4–13.7)	<0.001 ***	13.5 (7.1–19.9)	5.6 (2.2–9.0)	0.016 *

^$^ Community Child Hunger Identification Project (CCHIP) questionnaire [5]. 95th CI = 95th percent confidence interval. * *p* < 0.05; ** *p* < 0.01; *** *p* < 0.001: Significant relationship, Rao–Scott Chi-Square test. ^#^ Significant difference between GTG and WC during 1999 for question 8: “Do any of your children ever go to bed hungry because there is not enough money to buy food?”, Rao–Scott Chi-Square test, *p* < 0.05.

**Table 4 ijerph-19-01038-t004:** Comparison of households falling within different hunger scale categories ^$^ by domain of residence within Gauteng and the Western Cape.

Levels of Food Security	Gauteng Weighted Percentage (Standard Error)							
	1999				2018			
	Urban Formal (%, SE)	Urban Informal (%, SE)	Rural (%, SE)	Rao-Scott Chi-Square *p* Value	Urban Formal (%, SE)	Urban Informal (%, SE)	Rural (%, SE)	Rao-Scott Chi-Square *p* Value
*N*	234	103	62		384	159	174	
Food Secure Hunger score = 0	46.5 (7.4)	21.1 (3.6)	28.9 (3.7)	<0.001 ***	58.4 (4.6)	44.6 (8.5)	63.4 (8.7)	0.159
At risk Hunger score 1–4	19.4 (2.8)	23.9 (3.0)	29.0 (2.3)		22.2 (2.7)	25.7 (4.9)	21.0 (4.6)	
Food shortage Hunger score 5–8	34.1 (6.1)	55.0 (5.9)	42.1 (1.9)		19.4 (2.9)	29.6 (4.5)	15.6 (4.9)	
	Western Cape Weighted Percentage (Standard Error)							
*N*	231	35	76		277	157	158	
Food Secure Hunger score = 0	51.4 (9.3)	6.8 (4.6)	22.7 (4.1)	<0.001 ***	50.2 (5.3)	31.2 (5.8)	52.4 (4.4)	0.025 *
At risk Hunger score 1–4	28.4 (5.0)	11.5 (4.3)	39.0 (3.1)		28.5 (3.2)	33.1 (3.6)	27.0 (4.0)	
Food shortage Hunger score 5–8	20.2 (4.8)	81.8 (8.9)	38.3 (6.6)		21.3 (3.0)	35.7 (5.7)	20.6 (3.5)	

^$^ Community Child Hunger Identification Project (CCHIP) questionnaire [5]. * *p* < 0.05; *** *p* < 0.001: Significant relationship, Rao–Scott Chi-Square test. SE: Standard error.

**Table 5 ijerph-19-01038-t005:** Comparison of mean [95% CI] and median [95% CI] macronutrient intakes of children 1–<10-years-old between 1999 and 2018. for Gauteng and Western Cape, by age group and overall.

**Energy and Macronutrient Intakes**		**Gauteng**							
		**1999**				**2018**			
		**1–3 yrs**	**4–6 yrs**	**7–<10yrs**	**All**	**1–3yrs**	**4–6yrs**	**7–<10yrs**	**All**
*N*		230	129	40	399	288	264	165	717
Energy kJ	Mean 95% CI	3822 *** [3563–4081]	4932 ** [4548–5316]	5142 ** [4499–5785]	4560 *** [4215–4905]	4859 [4498–5220]	5867 [5511–6223]	6811 [6271–7351]	5724 [5520–5930]
	Median 95% CI	3665 &&& [3315–4014]	4792 && [4346–5239]	5009 && [3820–6198]	4479 &&& [4040–4917]	4306 [3731–4880]	5448 [5090–5806]	6386 [5957–6815]	5403 [5173–5633]
Total protein g	Mean 95% CI	28.9 * [26.9–30.8]	39.2 [36.1–42.4]	41.0 [35.1–46.8]	35.7 [33.0–38.4]	32.8 [30.0–35.5]	40.8 [37.6–43.9]	45.6 [41.8–49.4]	38.9 [37.2–40.6]
	Median 95%	25.5 && [22.8–28.2]	38.3 [34.5–42.2]	39.0 [32.3–45.6]	34.0 &&& [30.1–38.0]	30.4 [26.9–33.8]	39.9 [36.0–43.8]	41.5 [37.8–45.1]	37.5 [35.2–39.8]
Total fat g	Mean 95% CI	25.4 *** [23.2–27.5]	33.0 ** [28.5–37.5]	35.6 *** [27.3–43.8]	30.7 *** [27.0–34.4]	36.1 [32.6–39.6]	46.5 [41.4–51.6]	57.9 [51.5–64.4]	45.5 [43.0–47.9]
	Median 95% CI	22.1 &&& [19.5–24.7]	28.4 &&& [23.5–33.4]	28.7 &&& [18.9–38.5]	27.1 &&& [23.5–30.8]	31.0 [26.5–35.5]	40.5 [36.8–44.3]	53.5 [44.4–62.6]	40.3 [36.9–43.6]
Total carbohydrates g	Mean 95% CI	132.8 *** [123.3–142.4]	167.9 ** [155.9–180.0]	171.3 *** [155.0–187.7]	155.4 *** [145.6–165.2]	167.3 [154.8–179.9]	195.6 [183.2–208.1]	222.2 [204.4–239.9]	191.6 [183.3–200.0]
	Median 95% CI	125.0 &&& [113.8–136.2]	163.9 && [151.4–176.5]	179.9 && [158.5–201.2]	151.9 &&& [140.6–163.3]	149.1 [137.6–160.6]	183.1 [172.5–193.8]	211.3 [193.0–229.5]	180.4 [172.8–187.9]
		**Western Cape**							
		**1999**				**2018**			
		**1–3 yrs**	**4–6 yrs**	**7–<10 yrs**	**All**	**1–3 yrs**	**4–6 yrs**	**7–<10 yrs**	**All**
*N*		155	129	58	342	240	197	155	592
Energy kJ	Mean 95% CI	5261 [4899–5622]	6641 [6198–7084]	7227 [6543–7911]	6267 [5906–6629]	5378 [5012–5744]	6212 [5823–6601]	6662 [6197–7127]	6030 [5767–6293]
	Median 95% CI	4893 [4473–5312]	6413 & [5892–6934]	6518 [5687–7349]	5971 [5607–6335]	5117 [4640–5594]	6015 [5486–6544]	6209 [5783–6636]	5699 [5384–6014]
Total protein g	Mean 95% CI	41.9 [37.9–46.0]	50.1 [45.8–54.3]	56.2 [49.3–63.1]	48.5 [44.7–52.3]	41.7 [38.9–44.5]	47.1 [43.9–50.2]	50.4 [47.1–53.7]	46.0 [44.4–47.6]
	Median 95%	40.6 [34.3–46.9]	46.3 [42.0–50.6]	46.8 [39.4–54.1]	44.4 [40.6–48.2]	39.6 [35.4–43.8]	44.2 [40.1–48.4]	49.4 [44.0–54.8]	44.0 [41.4–46.6]
Total fat g	Mean 95% CI	43.4 [37.1–49.6]	53.5 [47.3–59.8]	57.6 [47.9–67.2]	50.7 [45.6–55.8]	43.6 [40.2–47.1]	53.8 [47.5–60.2]	62.8 [56.4–69.2]	52.6 [48.7–56.6]
	Median 95% CI	41.1 [34.7–47.5]	51.2 [43.2–59.2]	54.0 [45.7–62.4]	47.7 [42.5–52.8]	40.1 [36.0–44.1]	46.9 [38.8–55.0]	56.9 [53.0–60.7]	48.7 [44.4–52.9]
Total carbohydrates g	Mean 95% CI	163.8 [153.3–174.3]	210.5 [197.1–224.0]	229.6 ** [212.7–246.4]	197.7 [187.8–207.6]	173.6 [156.6–190.6]	197.6 [183.8–211.4]	198.8 [184.3–213.3]	189.0 [177.7–200.2]
	Median 95% CI	151.8 [134.1–169.5]	203.3 [182.0–224.7]	231.8 & [209.6–254.0]	188.7 [180.8–196.7]	155.5 [143.2–167.7]	180.7 [158.3–203.1]	182.9 [164.9–200.9]	170.9 [159.2–182.6]

CI = Confidence Interval; yrs = years; kJ = kilojoules; g = gram. * *p* < 0.05, ** *p*< 0.01, *** *p* < 0.001: Significant difference between macronutrient intake in 1999 and 2018 comparing by age group, independent *t*-test (using mean values). & *p* < 0.05, && *p* < 0.01, &&& *p* < 0.001: Significant difference between macronutrient intake in 1999 and 2018 comparing by age group, Wilcoxon two-sample test (using median values). Means calculated by incorporating weights and complex survey designs; CI = 95% confidence interval.

**Table 6 ijerph-19-01038-t006:** Quality (nutrient density) of the top 12 food items contributing to the total energy intake of 1–<10-year-old children within Gauteng and the Western Cape, in 1999 and 2018.

Rank	Gauteng 1999	% kJ	Gauteng 2018	% kJ
1	Maize porridge (not fortified) *	25.3	Maize porridge (fortified) *	26.1
2	Brown bread (not fortified)	9.8	Salty snacks (crisps, popcorn)	5.6
3	White bread (not fortified)	6.1	Brown bread (fortified)	4.2
4	Sugar	6.0	Potato with fat	4.1 ^b^
5	Whole milk	4.6	Sugar	4.0
6	Chicken (all types)	4.0	White bread (fortified)	3.9
7	Beef (all types)	3.4	Chicken (all types)	3.8
8	Hard margarine (saturated fat)	2.6	Poly-unsaturated oil	2.9
9	White rice	2.5	Whole milk	2.6
10	Eggs (any preparation)	2.4	White rice	2.6
11	Fresh fruit (other)	2.2	Sweets and chocolates (candy)	2.3
12	Vitamin C rich vegetables	2.0	Commercially processed meats	2.4
Rank	WC 2019	% kJ	WC 2018	% kJ
1	White bread (not fortified)	9.3	Maize porridge (fortified	16.0
2	Whole milk	6.9	Salty snacks	6.5
3	Sugar	5.7	White bread (fortified)	6.4
4	Hard margarine (saturated fat)	4.8	Chicken (all types)	6.1
5	Brown bread (not fortified)	4.6	Potato with fat	5.8 ^c^
6	Maize porridge (not fortified)	4.4	Whole milk	4.3
7	White rice	4.2	White rice	3.9
8	Plain beef	4.1	Commercially processed meats	3.6
9	Sugar sweetened cold drink	3.9	Sugar	3.5
10	Plain chicken	3.7	High fibre cereal	3.5
11	Fresh fruit (other)	3.5	Poly unsaturated fat (medium)	3.0
12	Potato with fat	3.4 ^a^	Pasta	2.7

Kj: Kilojoule; Colour key: Energy dense nutrient poor or moderate energy nutrient poor; Energy moderate/dense and nutrient rich; Nutrient dense (high micronutrients, low energy/g). Nutrient dense (high micronutrients, low energy/g); Fresh fruit (not vitamin C rich) such as apples, pears, plums, grapes; Vitamin C rich vegetables such as coleslaw, cabbage, tomatoes; * Fortification of white and brown bread and maize flour only became mandatory in 2003 [12].^a^ Of which 1.2% from ‘slap chips’ (French fries); ^b^ Of which 3% from ‘slap chips’; ^c^ Of which 3.5% from ‘slap chips’.

**Table 7 ijerph-19-01038-t007:** Comparison of prevalence of stunting, wasting, overweight and obesity in 1–<10-year-old children between 1999 and 2018 by age group and overall, within Gauteng and the Western Cape in 1999 and 2018.

Anthropometric Results	Gauteng Weighted Percentage (Standard Error)							
	1999				2018			
	1–3 yrs (%, SE)	4–6 yrs (%, SE)	7–<10 yrs (%, SE)	All (%, SE)	1–3 yrs (%, SE)	4–6 yrs (%, SE)	7–<10 yrs (%, SE)	All (%, SE)
*N*	219	119	48	386	281	262	164	707
Stunted: HAZ < −2SD	29.0 (2.0)	11.9 (2.8)	6.9 (3.3)	16.7 (1.9)	24.8 (3.7)	7.2 (2.0)	7.8 (2.6)	14.2 (1.9)
Wasted: BAZ < −2SD	9.3 (1.6)	14.0 (4.3)	6.9 * (3.5)	10.3 (1.9)	6.1 (2.2)	19.2 (3.8)	20.2 (3.7)	14.4 (2.1)
Overweight: BAZ > +2SD & ≤ 3SD (<5-year-olds) or > 1SD & ≤ 2SD (≥5-year-olds)	10.4 (2.2)	7.8 (2.7)	19.0 (6.1)	11.9 (2.0)	10.3 (2.2)	10.6 (3.0)	9.7 (2.3)	10.2 (1.4)
Obese: BAZ > +3SD (<5-year-olds) or > +2SD (≥5-year-olds)	1.7 * (1.2)	7.6 (2.4)	9.2 (5.0)	5.9 (1.7)	8.1 (1.8)	5.0 (1.6)	9.3 (3.0)	7.4 (1.4)
	Western Cape weighted percentage (standard error)							
*N*	134	119	56	309	231	193	152	576
Stunted: HAZ < −2SD	16.1 (2.5)	14.0 (2.9)	14.5 * (5.4)	14.9 (2.4)	24.9 (5.0)	8.9 (2.5)	4.9 (2.0)	13.7 (2.5)
Wasted: BAZ < −2SD:	12.1 (3.6)	7.7 (2.8)	18.1 (4.1)	11.8 (2.6)	9.2 (2.3)	8.4 (2.6)	13.2 (2.9)	10.1 (1.4)
Overweigh: BAZ > +2SD & ≤ 3SD (<5-year-olds) or > 1SD & ≤ 2SD (≥5-year-olds)	6.3 ** (2.1)	8.0 (3.1)	15.9 (5.2)	9.3 (1.8)	17.3 (3.8)	11.3 (3.0)	17.8 (3.6)	15.4 (1.9)
Obese: BAZ > +3SD (<5-year-olds) or > +2SD (≥5-year-olds):	2.1 (1.3)	3.3 (2.1)	4.4 (1.9)	4.2 (1.6)	6.9 (1.7)	8.4 (2.5)	3.1 (1.7)	6.4 (1.0)

* *p* < 0.05; ** *p* < 0.01: Rao–Scott Chi-Square test, weighted analysis with complex survey design; SE: Standard error.

**Table 8 ijerph-19-01038-t008:** Correlation between the hunger score and dietary and anthropometric indicators in 1–<10-year-old children within Gauteng and the Western Cape in 1999 and 2018.

Dietary and Anthropometric Variables	Gauteng		Western Cape	
	1999	2018	1999	2018
	*N* = 399	*N* = 717	*N* = 342	*N* = 592
Energy	−0.22 ***	0.03	−0.30 ***	−0.11 **
Total protein	−0.22 ***	−0.04	−0.32 ***	−0.15 **
Total fat	−0.27 ***	−0.08 *	−0.35 ***	−0.23 ***
Carbohydrates	−0.16 **	0.07	−0.20 **	−0.01
N (Z-scores)	*N* = 386	*N* = 707	*N* = 309	*N* = 576
HAZ	−0.14 *	−0.02	−0.34 ***	−0.11 *
WAZ	−0.15 **	−0.05	−0.32 ***	−0.07
BAZ	−0.07	−0.05	−0.15 **	−0.00

* *p* < 0.05; ** *p* < 0.01; *** *p* < 0.001: Test whether Spearman correlation with hunger score differs from zero.

**Table 9 ijerph-19-01038-t009:** Bivariate logistic regression to identify socio-demographic predictors of food shortage in 1–<10-year-old children in Gauteng and Western Cape, in 1999 and 2018.

Sociodemographic Variables	1999		2018	
	Gauteng Odds Ratio (95% CI)	Western Cape Odds Ratio (95% CI)	Gauteng Odds Ratio (95% CI)	Western Cape Odds Ratio (95% CI)
**Who looks after the child most of the time**	*N* = 418 (*n* = 172)	*N* = 353 (*n* = 111)	*N* = 733 (*n* = 151)	*N* = 593 (*n* = 150)
Mother/Father	Ref	Ref	Ref	Ref
Grandparent	1.80 (0.9 –3.28)	1.85 (0.59–5.80)	0.67 (0.31–1.47)	1.01 (0.59–1.74)
Other (sibling, aunt, uncle)	0.65 (0.20–2.14)	4.19 (1.21–14.49) *	0.15 (0.03–0.73) *	0.64 (0.25–1.64)
**Age of child**				
1–3 years	Ref	Ref	Ref	Ref
4–6 years	0.67 (0.50–0.90) *	0.75 (0.31–1.87)	1.51 (0.90–2.54)	1.01 (0.44–2.33)
7–<10 years	0.88 (0.46–1.66)	0.34 (0.16–0.69) **	1.05 (0.59–1.87)	0.94 (0.48–1.83)
**Gender**				
Male	Ref	Ref	Ref	Ref
Female	1.02 (0.60–1.72)	1.30 (0.71–2.39)	1.36 (0.86–2.15)	0.70 (0.41–1.18)
**Head of household**				
Father	Ref	Ref	Ref	Ref
Mother	1.69 (0.70–4.07)	1.73 (0.69–4.34)	1.31 (0.74–2.32)	1.76 (0.87–3.56)
Grandparent	2.00 (1.12–3.56) *	2.66 (0.98–7.22)	1.34 (0.82–2.17)	1.09 (0.61–1.95)
Other (e.g., aunt, uncle, friend)	1.02 (0.37–2.81)	6.72 (1.42–31.67) *	0.80 (0.26–2.45)	1.37 (0.48–3.91)
**Marital status of mother**				
Married	Ref	Ref	Ref	Ref
Other	2.58 (1.30–5.12) **	4.36 (2.00–9.47) **	1.70 (0.93–3.12)	1.80 (0.99–3.28)
**Mother’s highest education**				
Less than matric ^#^	Ref	Ref	Ref	Ref
Matric	0.35 (0.20–0.59) **	0.23 (0.11–0.49) **	0.24 (0.14–0.41) ***	0.51 (0.24–1.07)
Qualification after matric	0.05 (0.01–0.22) **	0.02 (0.002–0.19) **	0.60 (0.33–1.09)	0.26 (0.06–1.13)
**Father’s highest education**				
Less than matric	-	-	Ref	Ref
Matric	-	-	0.43 (0.20–0.90) *	0.47 (0.22–1.02)
Qualification after matric	-	-	0.46 (0.25–0.84) *	0.18 (0.06–0.61) **
**Mother’s employment status**				
No/don’t know/NA	Ref	Ref	Ref	Ref
Yes	0.29 (0.13–0.68) **	0.42 (0.22–0.79) *	0.28 (0.13–0.62) **	0.38 (0.19–0.75) **
**Father’s employment status**				
No/Don’t know/NA	Ref	Ref	Ref	Ref
Yes	0.66 (0.36–1.23)	0.29 (0.15–0.57) **	0.40 (0.26–0.63) **	0.32 (0.17–0.63) **
**Type of residence**				
Urban formal	Ref	Ref	Ref	Ref
Urban informal	2.22 (1.06–4.62) *	17.24 (4.04–73.51) **	1.77 (0.99–3.17)	2.07 (1.08–3.98) *
Rural	1.35 (0.72–2.50)	2.71 (1.06–6.94) *	0.78 (0.33–1.85)	0.97 (0.54–1.74)
**Year**	**% Shortage (SE)**	**% Shortage (SE)**	**Odds Ratio (95% CI)**	**Odds Ratio (95% CI)**
1999	-	-	Ref	Ref
2018	-	-	0.37 (0.22–0.60) **	0.71 (0.42–1.17)

Ref = reference value; CI = Confidence interval; The category “Matric” represents passed grade 11 or grade 12 in 1999, but only grade 12 in 2018. 95%CI, 95% confidence intervals. * *p* < 0.05; ** *p* < 0.01; *** *p* < 0.001: Significant odds ratio. N-values reflect actual number of cases, n values reflect the actual size of the risk group; estimates are adjusted using relevant weighting. Logistic regressions performed by incorporating weights and complex survey design. In the last entry (year), we compare “shortage of food” between the two surveys, using logistic regression.

## Data Availability

The data presented in this study are available on request from the corresponding author pending ethical approval from the Faculty of Health Sciences Human Research Ethics Committee, University of Cape Town.

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
