# Peer review of "Has Food Security and Nutritional Status Improved in Children 1–<10 Years in Two Provinces of South Africa between 1999 (National Food Consumption Survey) and 2018 (Provincial Dietary Intake Study (PDIS))"

_ijerph, 2022, doi:10.3390/ijerph19031038_

Round 1
Reviewer 1 Report
The idea and content are pretty good. However, the interpretation of the results should be carefully performed. For instance, the food security and nutritional status should include whether the freshness of the perishable foods improved. For instance, seafood spoilage will not change too much the calories but affected the quality significantly. eg., Food Control, 123, 107697;
Another key term should be discussed is whether more nutrient-dense foods were consumed in 2018. The low calorie does not necessarily poor but may due to some health concern. Thus another key aspect 'nutrient dense' foods should be discussed, fresh produce, seafood etc. For instance, Food Chemistry, 354, 129581.
Author Response
|
Reviewer 1 |
Thanks for valuable comments |
|
The idea and content are pretty good. However, the interpretation of the results should be carefully performed. For instance, the food security and nutritional status should include whether the freshness of the perishable foods improved. For instance, seafood spoilage will not change too much the calories but affected the quality significantly. eg., Food Control, 123, 107697; |
The main food items consumed in 1999 and 2018 remained similar with the main contributors to energy intake being maize porridge, bread, salty snacks and granulated sugar. We did not examine the perishability of food items since this was not an objective of the study and intake of fish, except canned fish, fruit and vegetables was low and did not contribute significantly to energy intake as you will notice in Table 6. |
|
Another key term should be discussed is whether more nutrient-dense foods were consumed in 2018. The low calorie does not necessarily poor but may due to some health concern. Thus another key aspect 'nutrient dense' foods should be discussed, fresh produce, seafood etc. For instance, Food Chemistry, 354, 129581. |
This was an important point made by the reviewer and we have subsequently added results on energy density. See table 6 and Lines 406-423. Energy intake increased significantly in GTG which explains some of the improvement in growth. However, many of the commonly consumed items are energy-dense, such as sugar sweetened beverages, candies and salty snacks [25]. |
Reviewer 2 Report
1) the novelty of this paper is not clearly articulated. The result may be produced by any government report, but what is the value addition of this scientific work?
2) Results from the socio-economic factors need to be highlighted broadly in the abstract and also in the conclusion.
Author Response
|
Reviewer 2 |
Thank you for your valuable comments |
|
1) the novelty of this paper is not clearly articulated. The result may be produced by any government report, but what is the value addition of this scientific work?
|
Thank you we agree with this comment and have subsequently refocused the aim of the study: lines 88-93 The aim of the present study was to determine whether food security and nutritional status (diet and anthropometry) have improved in households of 1 - <10-year-old children in two provinces in since the 1999 NFCS. Furthermore, we aimed to investigate sociodemographic profiles of the households, sociodemographic predictors of food shortage in these households and relationships between household food security and child dietary and malnutrition indicators at each time point. In order to explain the changes in food security in the two provinces we tried to evaluate all the available data to account for these changes. In order to do this, we also added additional data, namely sociodemographic factors at the two time points (Table 1), dietary changes (Table 6) and examined the association between food security and diet and anthropometry (Table 8) at the two time points. In the discussion we also described some additional factors which may have led to changes such as government initiatives introduced after 1994 when South Africa became a democracy.
|
|
2) Results from the socio-economic factors need to be highlighted broadly in the abstract and also in the conclusion |
Table 1 with the socio-demographic factors has been added and in abstract lines 28-30 reflect this. In results section lines 459-475; in discussion lines 491-496; 614-617; 622-627 and in conclusion 641-644. |
Reviewer 3 Report
abstract
Line 14 -23 - reorganize percentages to be in the same order as corresponding dates - from 55% to to 29.6%. Also use consistent number of decimal places.
Introduction
Line 40 - states 1999 study was on children aged 1-9 so how can comparisons be made for children older than that as stated in title?
Line 73 - does not make sense
Methods
Line 113 - repeated sentence
Line 117 - the calculation use to obtain sample size is not clear.
Results
Throughout all charts ensure what is actually being compared to obtain significance is clearly explained.
Table 1: it is very confusing to put p values in one column that pertain to various columns. Reorganize the table to put the p value clearing next to relevant result.
Line 305-308 this is a repeated line
Table 5 explanation cut off so not sure what * ** mean
Discussion
Line 387-390 : clarify years and locations you are talking about
Is it not also important to note that wasting actually increased among older children in Gauteng and discuss why this could be happening?
Author Response
|
Reviewer 3 |
Thank you for your valuable comments |
|
abstract Line 14 -23 - reorganize percentages to be in the same order as corresponding dates - from 55% to to 29.6%. Also use consistent number of decimal places.
|
This has been corrected |
|
Line 40 - states 1999 study was on children aged 1-9 so how can comparisons be made for children older than that as stated in title?
|
The title had a typing error and we have corrected it |
|
Introduction Line 73 - does not make sense
|
Corrected lines 77-79 Improvements have also been made to health services with the introduction of the Integrated School Health Program (ISHP)[13] comprising three core components, namely: health assessment and screening, on-site health services, and health promotion. Additionally, improved primary health care clinic services and free health services for children between 0-6 years and expectant and breastfeeding mothers were introduced [14] |
|
Methods Line 113 - repeated sentence
|
Corrected |
|
Line 117 - the calculation use to obtain sample size is not clear |
Lines 127-133. The sampling was done according to the USAID Household Sampling and Listing Method. This is part of the DHS survey toolkit.[16]y |
|
Table 1: it is very confusing to put p values in one column that pertain to various columns. Reorganize the table to put the p value clearing next to relevant result.
|
This has been corrected in what is now Table 2 |
|
Line 305-308 this is a repeated line
|
The repetition has been deleted |
|
Table 5 explanation cut off so not sure what * ** means
|
This has been corrected under Table 5. |
|
Discussion Line 387-390 : clarify years and locations you are talking about
|
This refers to all the children studied in both provinces (lines 485-487) |
|
Is it not also important to note that wasting actually increased among older children in Gauteng and discuss why this could be happening?
|
We agree that this needs to be mentioned. We have added this to lines 21-22 in abstract, lines 435-437 in results and lines 510-512 in discussion Overall wasting increased from 10.3% to 14.4% in GTG and decreased from 11.8 to 10.1% in WC. It is particularly high in older children (6-<10 years) and is difficult to explain since primary school children receive school meals. |
|
|
|
|
|
|
|
Reviewer 3 |
Thank you for your valuable comments |
|
abstract Line 14 -23 - reorganize percentages to be in the same order as corresponding dates - from 55% to to 29.6%. Also use consistent number of decimal places.
|
This has been corrected |
|
Line 40 - states 1999 study was on children aged 1-9 so how can comparisons be made for children older than that as stated in title?
|
The title had a typing error and we have corrected it |
|
Introduction Line 73 - does not make sense
|
Corrected lines 77-79 Improvements have also been made to health services with the introduction of the Integrated School Health Program (ISHP)[13] comprising three core components, namely: health assessment and screening, on-site health services, and health promotion. Additionally, improved primary health care clinic services and free health services for children between 0-6 years and expectant and breastfeeding mothers were introduced [14] |
|
Methods Line 113 - repeated sentence
|
Corrected |
|
Line 117 - the calculation use to obtain sample size is not clear |
Lines 127-133. The sampling was done according to the USAID Household Sampling and Listing Method. This is part of the DHS survey toolkit.[16]y |
|
Table 1: it is very confusing to put p values in one column that pertain to various columns. Reorganize the table to put the p value clearing next to relevant result.
|
This has been corrected in what is now Table 2 |
|
Line 305-308 this is a repeated line
|
The repetition has been deleted |
|
Table 5 explanation cut off so not sure what * ** means
|
This has been corrected under Table 5. |
|
Discussion Line 387-390 : clarify years and locations you are talking about
|
This refers to all the children studied in both provinces (lines 485-487) |
|
Is it not also important to note that wasting actually increased among older children in Gauteng and discuss why this could be happening?
|
We agree that this needs to be mentioned. We have added this to lines 21-22 in abstract, lines 435-437 in results and lines 510-512 in discussion Overall wasting increased from 10.3% to 14.4% in GTG and decreased from 11.8 to 10.1% in WC. It is particularly high in older children (6-<10 years) and is difficult to explain since primary school children receive school meals. |
|
|
|
|
|
|
Reviewer 4 Report
The topic that the paper describes it is important. However, I think that the quality of the paper should be improved extensively. It seems that you mention some data and present a report of your results but youu do not tell a stody. Moreover, what is the importance of this paper from a scientific point of view? You start by mentioning some data about hunger levels in the introduction, present the results and then discuss about grants. What is the connection between all these things? Thus, keep in mind to tell the story. Finally, it is not very clear what is the relevance of your paper. Why scientists should read it? What is the gap that is trying to fill? You have not mentioned what the literature on this topic is about.
The title seems to start as a question but i dont see the question mark.
Lines 14 to 17 not very clear. Try to devide the sentence
Introduction:
- This part is a bit confusing. The paper seems to be for children, but you also mention adults. The paragraphs seems disconnected. Probably you should start by mentioning the general conditions of food security and then explain how it is reflected to children. Remember, you are telling a story.
- line 33, in children should be written among children
- line 34: You start talking about children and then you mentioned women 15-49 years old. Even though you might have mentioned it to illustrate the situation, I find it confusing. Stick to children.
In the methodology's part, it is not very clear if the regions are the same for both, 1999 and 2018. Explain this.
Conclusion: Here you are talking extensively about grants and their impacts but you do not mention results of your study. Moreover, in the introduction you do not mention grants. Subsequently, it is not very clear from where grants come out. Remember that in this section you should explain the implications that your results have.
Author Response
|
Reviewer 4 |
Thank you for your valuable comments |
|
The topic that the paper describes it is important. However, I think that the quality of the paper should be improved extensively. It seems that you mention some data and present a report of your results but you do not tell a story. Moreover, what is the importance of this paper from a scientific point of view? You start by mentioning some data about hunger levels in the introduction, present the results and then discuss about grants. What is the connection between all these things? Thus, keep in mind to tell the story. Finally, it is not very clear what is the relevance of your paper. Why scientists should read it? What is the gap that is trying to fill? You have not mentioned what the literature on this topic is about.
|
Thank you we agree with this comment and have subsequently refocused the aim of the study: lines 88-93 The aim of the present study was to determine whether food security and nutritional status (diet and anthropometry) have improved in households of 1 - <10-year-old children in two provinces in since the 1999 NFCS. Furthermore, we aimed to investigate sociodemographic profiles of the households, sociodemographic predictors of food shortage in these households and relationships between household food security and child dietary and malnutrition indicators at each time point. In order to explain the changes in food security in the two provinces we tried to evaluate all the available data to account for these changes. In order to do this, we also added additional data, namely sociodemographic factors at the two time points (Table 1), dietary changes (Table 6) and examined the association between food security and diet and anthropometry (Table 8) at the two time points. In the discussion we also described some additional factors which may have led to changes such as government initiatives introduced after 1994 when South Africa became a democracy.
|
|
The title seems to start as a question but I dont see the question mark.
|
Question mark has been added |
|
Lines 14 to 17 not very clear. Try to divide the sentence
|
Corrected as requested also by another reviewer. |
|
Introduction:
|
1. Unfortunately there was a typing error in the title which has been corrected to children under 10 years old.
2. Correction made 3. The statistics on overweight/obesity in South African women were included to illustrate the finding that the double burden of malnutrition exists in South Africa. However, the reviewer makes an important point hence we have removed this sentence
|
|
In the methodology's part, it is not very clear if the regions are the same for both, 1999 and 2018. Explain this.
|
Random EAs were drawn in 1999 and 2018 keeping the domains the same, namely urban formal, urban informal and rural EAs. Every attempt was made to keep the repeated survey as similar as possible to that of 1999. |
|
Conclusion: Here you are talking extensively about grants and their impacts but you do not mention results of your study. Moreover, in the introduction you do not mention grants. Subsequently, it is not very clear from where grants come out. Remember that in this section you should explain the implications that your results have.
|
Grants are mentioned in the introduction in lines 71-81: To combat poverty and food insecurity the South African Government has introduced numerous social benefits and programs, including a social grants (cash transfers) such as the childcare and foster care grants [10]. Furthermore, a National School Nutrition Program (NSNP) has been introduced to provide hungry children with a daily meal at school [11], community food garden initiatives and a national fortification program was introduced to combat micronutrient deficiencies [12]. Improvements have also been made to health services with the introduction of the Integrated School Health Program (ISHP)[13] comprising three core components, namely: health assessment and screening, on-site health services, and health promotion. Additionally, improved primary health care clinic services and free health services for children between 0-6 years and expectant and breastfeeding mothers [14].
We have also refocused the aim: The aim of the present study was to determine whether food security and nutritional status (diet and anthropometry) have improved in households of 1 - <10-year-old children in two provinces in since the 1999 NFCS. Furthermore, we aimed to investigate sociodemographic profiles of the households, sociodemographic predictors of food shortage in these households and relationships between household food security and child dietary and malnutrition indicators at each time point.
In order to explain the changes in food security in the two provinces we tried to evaluate all the available data to account for these changes. In order to do this, we have added additional data, namely sociodemographic factors at the two time points (Table 1), dietary changes (Table 6) and examined the association between food security and diet and anthropometry (Table 8) at the two time points. In the discussion we also described some additional factors which may have led to changes such as government initiatives introduced after 1994 when South Africa became a democracy.
Conclusion: In contrast with 1999, no predictors of risk of food security remained in GTG and only one in the WC (living in an informal area). Protectors against food shortage remained similar over time and mostly involved having better educated and employed parents. The improvement in food security may have contributed to the increase in energy intake of children in GTG and subsequent improvements in nutritional status. However, it is a concern that the improvement in food security has been accompanied by increased overweight and obesity, especially in the younger children in both provinces. It is not possible to say conclusively that the government initiatives led to the improvements in food security but evidence from the South African literature has indicated that many of these initiatives have been successful, particularly the child support grant and the NSNP.
|
|
|
|
Reviewer 5 Report
This is an interesting paper with a great deal of data. The authors conducted adequate analysis and conclusions are supported by the data. Methods used are simple but sound. I have only minor comments regarding the paper.
- The tile talks about "children 1-<20years" but the paper present data about children 1-10 years.
- Table 1 could have separate columns for the p-values, just as table 2. It would be easier to understand.
- Table 4 "gauteng" panel shows "total protein g" twice.
Author Response
|
Reviewer 5 |
Thank you for your valuable comments |
|
This is an interesting paper with a great deal of data. The authors conducted adequate analysis and conclusions are supported by the data. Methods used are simple but sound. I have only minor comments regarding the paper.
|
|
|
1.The title had a typing error which has been corrected to children <10 years
2. Table 1 (now 2) has been simplified.
3. This has been corrected. |
Round 2
Reviewer 3 Report
Abstract
Line 21: unclear what the comparison group is: The 7 - <10-year-olds (6.9 versus 20.2%) in GTG were significantly more likely to be wasted in 2018, and the 1 – 3-year-olds more likely to be obese (1.7% versus 8.1%). In the WC 7 - <10-year-olds were significantly less likely to be stunted in 2018 (14.5% versus 4.9%) and 1 – 3- year-olds more likely to be overweight (6.3% versus 17.3%).
Line 24: need to clarify to get to main points. There were significant negative correlations between the hunger score and dietary variables in both provinces in WC 1999. In GTG only the correlation with fat intake remained. Changes in top 12 energy contributors reflect a shift to high or moderate foods low in nutrients from 1999 to 2018. Logistic regression analyses reflect the importance of having a parent as head of the household and/or caregiver, and parents having grade 12 or higher education and being employed in ensuring food security.
Method
Line 129 repeated line still there
Results:
Line 298: 50% of what?
Table 1: in table not clear which numbers are n or %. Also looks like only p values for comparison within same province at different time points are included so where is the comparison between provinces as per the title?
Table 2: unclear how comparisons are being shown in this table.
Line 428: which province?
Table 9: unclear what is being compared (highlighted) by the, , ***. Some things mentioned in the text are not highlighted in any way so confusing.
Review all tables to ensure the reader does not need to spend excessive time figuring out what is compared.
Author Response
Reviewer 3. Thank you for detailed comments and for pointing out errors in the text.
Line 21: unclear what the comparison group is: The 7 - <10-year-olds (6.9 versus 20.2%) in GTG were significantly more likely to be wasted in 2018, and the 1 – 3-year-olds more likely to be obese (1.7% versus 8.1%). In the WC 7 - <10-year-olds were significantly less likely to be stunted in 2018 (14.5% versus 4.9%) and 1 – 3- year-olds more likely to be overweight (6.3% versus 17.3%).
Replaced by:
The only significant change in stunting, wasting, overweight and obesity prevalence was that 7 - <10-year-olds in GTG were significantly more likely to be wasted (BAZ<-2SD) in 2018 than in 1999 (20.2% versus 6.9% respectively). In the WC 1– 3-year-olds were significantly more likely to be obese in 2018 than in 1999 (8.1% versus 1.7% respectively) and 7 - <10-year-olds were less likely to be stunted (14.5% versus 4.9% respectively).
Line 24: need to clarify to get to main points. There were significant negative correlations between the hunger score and dietary variables in both provinces in WC 1999. In GTG only the correlation with fat intake remained.
Replaced by: There were significant negative correlations between the hunger score and dietary variables in both provinces in 1999. In GTG in 2018 only the correlation with fat intake remained while there were still several significant correlations in WC in 2018.
Changes in top 12 energy contributors reflect a shift to high or moderate foods low in nutrients from 1999 to 2018. Nutrient dense (high micronutrients, low energy/g) foods (e.g. fruit) fell off the list in 2018.
Logistic regression analyses reflect the importance of having a parent as head of the household and/or caregiver, and parents having grade 12 or higher education and being employed in ensuring food security. This has not changed since it reflects the main outcomes of the regression
Method
Line 129 repeated line still there
Apology for missing that. It has now been deleted
Results:
Line 298: 50% of what?
for the mother’s highest level of education (just under or just more than 50% in both provinces at both time points) had an education level less than matric.(now line 309)
Table 1: in table not clear which numbers are n or %. Also looks like only p values for comparison within same province at different time points are included so where is the comparison between provinces as per the title?
At the top of table is shown weighted percentage (standard error). Hence the first column is a % and the standard error is in brackets. Between provinces have also been compared. See below table:
*Rao-Scott Chi-Square test; comparison within each province in 1999 and 2018 values of sociodemographic variables were adjusted using relevant weighting. Frequencies performed by incorporating weights and complex survey design. *p<0.05; **p<0.01; ***p<0.001
2 Rao-Scott Chi-Square test; comparison between provinces in 1999; relationships: marital status of mother (p<0.05) & employment status of mother (p<0.001). Not shown in the table (these were the only significant outcomes)
3 Rao-Scott Chi-Square test; comparison between provinces in 2018; relationships: marital status of mother (p<0.001) & employment status of mother (p<0.001). Not shown in the table (these were the only significant outcomes)
Table 2: unclear how comparisons are being shown in this table.
*p<0.05; **p<0.01; ***p<0.001: Significant relationships within provinces comparing 1999 with 2018, Rao-Scott Chi-Square test. Means calculated by incorporating weights and complex survey designs; CI=95% confidence interval
#p<0.05; ###p<0.001: Significant difference within provinces of GTG and WC mean hunger scores in 1999 and 2018, Wilcoxon two-sample test.
Line 428: which province? Added to the text: in GTG (now line 444)
Table 9: unclear what is being compared (highlighted) by the, ***. Some things mentioned in the text are not highlighted in any way so confusing.
The stars represent different levels of significance of the odds ratio as indicated under the table.
. *p<0.05; **p<0.01; ***p<0.001: Significant odds ratio
We have checked that everything in the table corresponds with that of the text.
Review all tables to ensure the reader does not need to spend excessive time figuring out what is compared.
We have checked every table
Reviewer 4 Report
I think that the quality of the paper has improved. However, there are some overall points that I think should still be improved. The introduction has improve a lot and there is a fluidity when reading it. However, I dont find a clear statement about the importance of this paper. Why is this paper important and how the results should be considered by the policy makers? Try to add a paragraph about this in the section of introduction, after the objective. Recall this as a concluding remark at the end of the paper. This is important since from a methodological point of view the paper is not scientifically powerful but its strength relies on its results and how they should be used from policymakers.Author Response
Reviewer 4. Thank you for your encouraging and inciteful comments. (See blue highlights in introduction and conclusion)
I think that the quality of the paper has improved. However, there are some overall points that I think should still be improved. The introduction has improve a lot and there is a fluidity when reading it. However, I dont find a clear statement about the importance of this paper. Why is this paper important and how the results should be considered by the policy makers? Try to add a paragraph about this in the section of introduction, after the objective. Recall this as a concluding remark at the end of the paper. This is important since from a methodological point of view the paper is not scientifically powerful but its strength relies on its results and how they should be used from policymakers.
Importance of the paper:
Introduction added:
The study has important implications for policy makers as it provides novel insights in whether food security and nutritional status of 1 to younger than 10-year-old children have improved after introduction of the mentioned initiatives since 1999. This is the only study that has been repeated in the same age group using the same methods over nearly two decades. Results will contribute to identification of the need for further improvement in food security in the Western Cape and Gauteng.
Conclusion added:
We conclude that the results of this study show that overall, the food security and anthropometry of 1-<10-year-old children in the WC and GTG improved since 1999. In contrast with 1999, no predictors of risk of food security remained in GTG and only one in the WC (living in an informal area). Protectors against food shortage remained similar over time and mostly involved having better educated and employed parents. The improvement in food security may have contributed to the increase in energy intake of children in GTG and subsequent improvements in nutritional status. However, it is a concern that the improvement in food security have been accompanied by increased overweight and obesity, especially in the younger children in both provinces.
In terms of policy formulation the results of this research imply that the various initiatives introduced by the South African government may have had the necessary effect. However, further research is essential to confirm the contribution and-cost-effectiveness of the different initiatives to identify those that should be continued into the future and those that should/could be phased out. The shift seen in food choices to more energy dense low nutrient items is a concern and should be taken up as a priority for intervention initaitives by local and national departments of health in the country.